# Near-term pathways for decarbonizing global concrete production

Josefine A. Olsson [1], Sabbie A. Miller [1] ✉ & Mark G. Alexander [2]

Growing urban populations and deteriorating infrastructure are driving unprecedented demands for concrete, a material for which there is no alternative that can meet its functional capacity. The production of concrete, more particularly the hydraulic cement that glues the material together, is one of the world's largest sources of greenhouse gas (GHG) emissions. While this is a well-studied source of emissions, the consequences of efficient structural design decisions on mitigating these emissions are not yet well known. Here, we show that a combination of manufacturing and engineering decisions have the potential to reduce over 76% of the GHG emissions from cement and concrete production, equivalent to 3.6 Gt $CO_2$-eq lower emissions in 2100. The studied methods similarly result in more efficient utilization of resources by lowering cement demand by up to 65%, leading to an expected reduction in all other environmental burdens. These findings show that the flexibility within current concrete design approaches can contribute to climate mitigation without requiring heavy capital investment in alternative manufacturing methods or alternative materials.

Cement-based materials are essential for urban development, and there is no alternative material that meets their functional capacity[1,2]. There are several uses of cement in such materials, such as in concrete and mortar (all composite materials using cement are referred to herein as concrete, which is its most common application). As the world population grows, the development, maintenance, and extension of urban areas will grow; projected estimates show that by 2030, nearly 1 billion (22% increase compared to 2018) more people will live in urban areas[3]. With such urban growth, the demand for concrete will continue to rise, with rates exceeding those of population growth[4].

Concrete is uniquely poised to meet the needs for many civil infrastructure and building systems because of the broad availability of the primary constituents of concrete, and the strength and durability achievable with this material[1,2]. Concrete consists of fine and coarse aggregates (sand and crushed rocks), water, admixtures, and a hydraulic binder (cement) that reacts with the water to glue these constituents together into an artificial conglomerate. Significant greenhouse gas (GHG) emissions are attributable to cement-based materials production, -8% of global anthropogenic $CO_2$ emissions[5],

which is primarily a function of producing clinker (the precursor to cement). Clinker is a calcined and quenched material that requires high temperatures to create the desired mineralogy, leading to emissions associated with fuels for thermal energy, and chemical-$CO_2$ emissions from limestone decarbonation in its production.

Society must reach net-zero GHG emissions by 2050 to limit warming to 1.5 °C above pre-industrial levels[6], and to do so, the "difficult-to-decarbonize" industries, such as cement and concrete[7], must find pathways to mitigation. There are several commonly discussed mitigation strategies for these emissions including use of alternative fuels, use of more efficient equipment, carbon capture, utilization and storage (CCUS), or reducing the demand for clinker through use of supplementary cementitious materials (SCMs)[8,9]. CCUS technologies are not well established for the industry[10], and while alternative cements and aggregates have been proposed[11–13], their efficacy can be hindered by resource availability, by costs, or by a risk-averse industry[14,15]. Critically, improving material efficiency, in which less material is used to achieve the same performance, is a key step in mitigating the environmental impacts from materials production[16–18]. This step should be used

[1]Department of Civil and Environmental Engineering, University of California, Davis, Davis, CA, USA. [2]Department of Civil Engineering, University of Cape Town, Cape Town, South Africa. ✉e-mail: sabmil@ucdavis.edu

in unison with low-emissions material alternatives to overcome GHG emissions challenges from the built environment.

Reducing material demand while meeting performance requirements will support provision of necessary infrastructure and contribute to reducing multiple environmental impacts. Yet, the role of engineering structural design in efficient use of concrete systems has been examined only limitedly[19,20]. In this work, we systematically quantify the potential role of manufacturing changes, in combination with mixture proportioning and engineering design in the efficient utilization of concrete worldwide (see Fig. 1). Herein, we consider emissions reductions resulting from: (i) the use of manufacturing changes with the potential to lower GHG emissions; (ii) changes to concrete mixture constituents and proportioning (e.g., reducing cement content through partial replacement with SCMs); (iii) variation in concrete compressive strength selected, reinforcement ratio selected, and design code implemented for reinforced concrete members; and (iv) the effect of increasing service life of buildings and infrastructure. These multiple methods are integrated to determine the cumulative effect of emissions reductions, some of which have been established or studied in isolation, such as Habert[9], Reis[21], Eleftheriadis[20], and Marsh[22]. The implications are critical in understanding how to drive alternative material technologies that will lead to significant GHG emissions mitigations, and policies that will guide appropriate application and use of concrete to meet societal demands, while mitigating emissions. Though reductions in GHG emissions can also be achieved by use of alternative structural systems, such as steel frames instead of concrete frames in certain scenarios, this study focuses only on mitigation from reinforced concrete. Analysis of reinforced concrete versus steel frames is highly dependent on the structural system under consideration and the results vary from case to case[23].

## Results

To show the opportunity for reduction of GHG emissions within current accepted design, we estimated the average global cradle-to-gate emissions from production of cement-based materials (concrete and mortar), based on 2015 baseline production values. A projection of global emissions from production of cement-based materials between 2015–2100 was modeled based on projected per capita saturation levels, projected population, and average in-use service lives of cement for 10 global regions. The model used was initially developed by Cao et al.[24].

The mitigation potentials for the strategies herein were estimated cumulatively, with reductions associated with manufacturing, design, and mixture proportioning increasing linearly between 2015–2100; benefits from increasing material longevity were modeled based on the dynamic effects for various concrete applications and the associated reduction in cement production in future years. The impact of improved manufacturing efficiency of cement and the effect of increasing global cement substitution by SCMs were determined as reduction of GHG emissions for a cubic meter of concrete compared to current production, and they were subsequently scaled to global concrete production to examine the global reduction in emissions. The potential reductions from optimizing structural design were calculated based on a relationship between mixture proportions (emissions from concrete production) and compressive strength[25] and the quantity of steel reinforcement used. Noting that the strength of concrete and the magnitude of steel reinforcement used will affect the volume of each of these materials that must be specified. To address this factor, a model linking concrete strength and reinforcement ratio to environmental impacts of a column or slab[26], for a unit structural frame, and based on three different design codes was derived. The impact of service life extension to reduce the future demand for cement was based on average in-use times of cement-based buildings and infrastructure in each of the 10 global regions.

### Changes in cement manufacturing and concrete production

To understand the efficacy of structural design in contributing to GHG emissions reductions, this work draws comparisons to the more conventional mitigation methods discussed, which typically revolve around manufacturing improvements for cement and concrete. While there are a variety of manufacturing improvements that can be implemented to reduce GHG emissions from cement and concrete production, here we consider common methods of improving the efficiency of cement kilns, replacement of high emitting kiln fuels with natural gas, and using a low-emissions electricity grid throughout the production of primary constituents (e.g., cement, aggregates).

To perform such a comparison, we derive GHG emissions associated with the production of concrete using established data reflecting current global practice as a baseline, and then assess the effects of using manufacturing alterations to lower these emissions. We consider process-based (e.g., emissions from limestone decarbonation) and energy-based emissions (e.g., from thermal energy resources, electricity demand, and transportation). While we address the role of

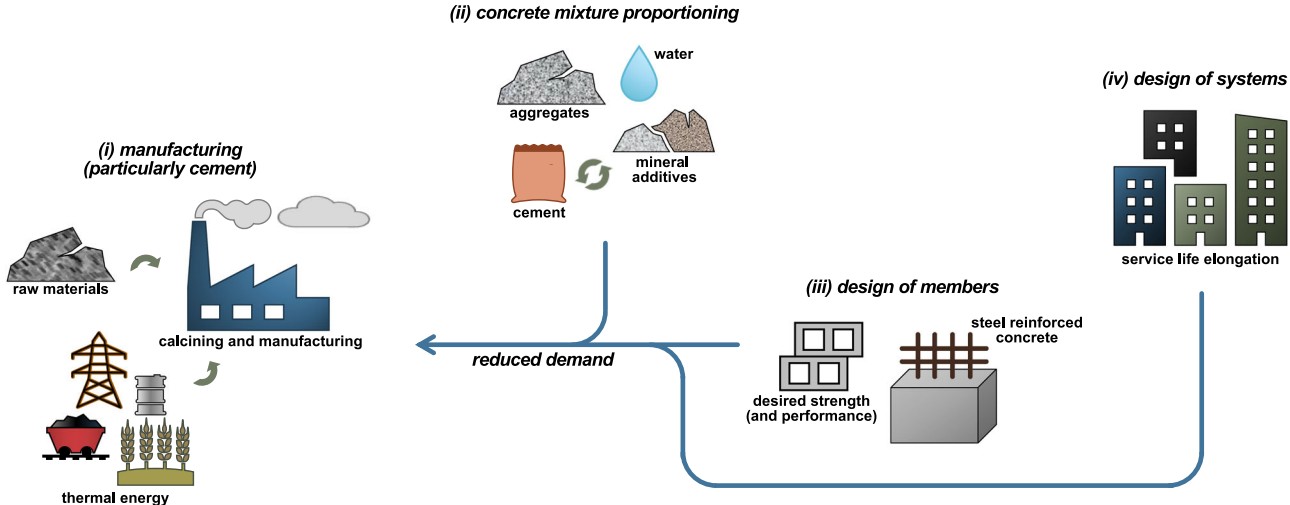

**Fig. 1 | Manufacturing and material efficiency improvements considered in this work.** Methods to reduce the GHG emissions in three design phases were studied to assess mitigation potential: (i) interventions at cement manufacture; (ii) interventions at concrete mixture proportioning; (iii) interventions at concrete member design – accounting for differences in global design standards; and (iv) interventions at built systems design and system service life extension. To ascertain the benefits of these phases, impacts are scaled to global mitigation potential.

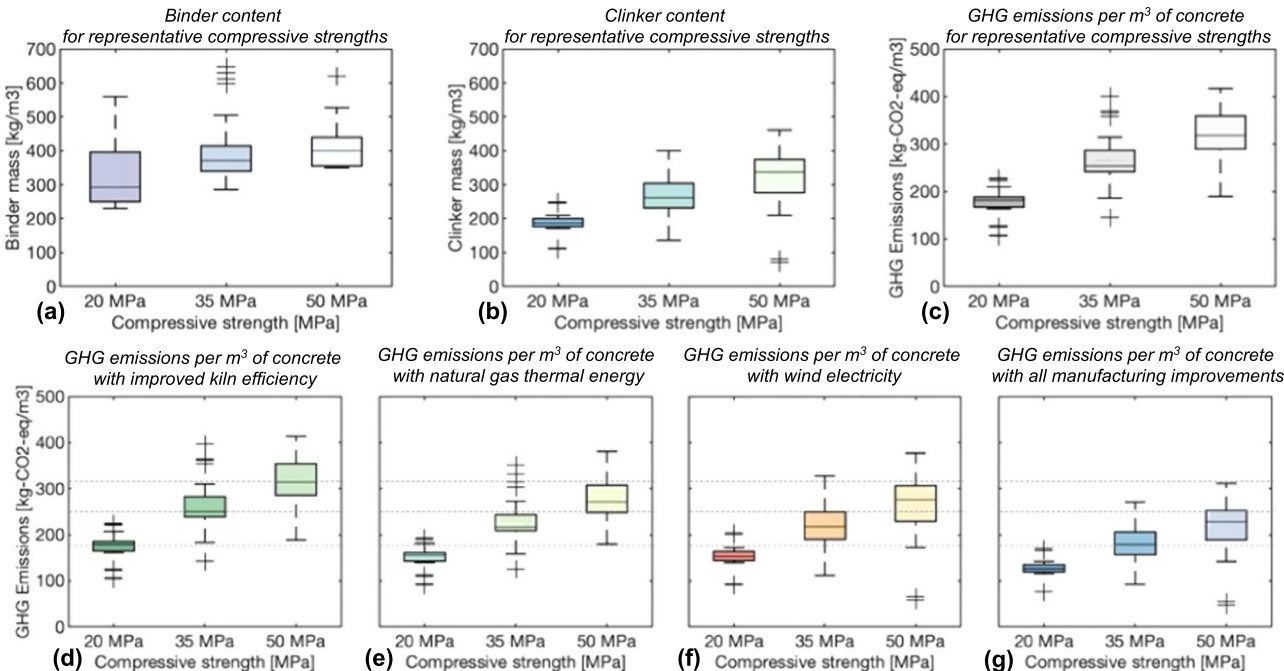

**Fig. 2 | Effects of binders, strength, and manufacturing improvements on greenhouse gas (GHG) emissions per m³ of concrete.** The correlation between clinker content, GHG emissions and compressive strength is confirmed for this selection of concrete mixtures, and it is shown that by implementing all below listed manufacturing improvements, the average GHG emissions per cubic meter of concrete can be reduced by ~20%. Additionally, among mixtures within the same strength class, there is a great variation in cement content and GHG emissions. **a** Example variation in binder content (here, we consider the binder to be the dry mass of clinker plus all mineral additives) for each of three compressive strength categories based on mixtures from the literature (see "Methods"); Portland cement and supplementary cementitious materials considered in the powder binder. **b** Example variation in clinker content for each of three compressive strengths.

**c** Approximate GHG emissions to produce these concrete mixtures. **d** GHG emissions from producing these concrete mixtures if kilns were all operating at the most efficient levels reported globally; note, the majority of current kilns are efficient, so little improvement is noted. **e** GHG emissions from producing these concrete mixtures if gas was used as to replace higher emitting thermal energy sources in kilns. **f** GHG emissions from producing these concrete mixtures if electricity demands in the supply chain were met through use of wind turbines. **g** GHG emissions from producing these concrete mixtures if all improvements (i.e., efficient kilns, gas thermal energy, and wind electricity) were used concurrently. Note: for (**d**–**g**) dashed lines represent mean emissions with no manufacturing improvements.

design decisions and use phase in subsequent stages, initial modeling is for cradle-to-gate impacts (i.e., from raw material acquisition through concrete batching) for one cubic meter of concrete. Assumptions for energy resources, initial levels of supplementary cementitious material use, and mixture proportions are stipulated in the Methods and the Supplementary Information.

Our results, which are in line with the GCCA roadmap[27], indicate that the common manufacturing alterations considered could contribute to GHG emissions reductions of 1% with increased kiln efficiency, ~15% with replacement of higher emitting thermal energy sources in kilns with natural gas, ~6% with wind electricity used to meet all electricity demands, thus ~20% for all manufacturing improvements combined (see Fig. 2d, e, f – data in Supplementary Data 1). Notably, in this work, we focus on measures that can be readily implemented, so we exclude technologies that are not currently established (e.g., CCUS). Further, we note the measures we present have been established as feasible (e.g., the wind electricity system for cement production in California's Mojave Desert), but similar emissions reductions could be achieved with other established technologies (e.g., solar electricity).

However, due to the common means to increase strength (i.e., higher cement content), there tends to be greater GHG emissions for higher strength concrete mixtures: for example, without any manufacturing improvements, there is a 75% increase in emissions for a 50 MPa mix relative to a 20 MPa mix (namely, a difference of 140 kg $CO_2$-eq/m³ between the median emissions per m³); a similar difference in median emissions remains even when the manufacturing improvements are implemented (130 kg $CO_2$-eq/m³) (see Fig. 2c – data in Supplementary Data 1).

However, just by selecting appropriate mixtures, such as those capable of achieving the desired strength with lower clinker content[28], emissions can be reduced in similar ranges without any manufacturing improvements. There is variability in GHG emissions within strength groups resulting from varying mixture proportions: from our data, there is ~20% variation at 20 MPa, ~40% variation at 35 MPa, and ~55% variation at 50 MPa, with a high correlation between concrete GHG emissions per m³ and clinker content within the mixture ($R^2 = 0.98$). Comparing the 25th percentile of GHG emissions to the median for concrete mixtures capable of achieving the same strength (with no manufacturing improvements), the 25th percentile is 8% lower than the median for 20 MPa, 4.5% lower than the median for 35 MPa, and 9% lower than the median for 50 MPa. These results could have substantial impacts on how we consider prescriptive design (where a minimum cementitious content is specified instead of a performance indicator, such as meeting a certain strength by 28 days). However, variability in GHG emissions per 28-day strength does not reflect other changes in performance characteristics obtained by cement replacement to achieve lower clinker content, such as durability and workability, which are important metrics in concrete mixture design.

## Structural design
Beyond the benefits gained from selecting desired mixture proportions, which can be implemented in many cases without capital investment in improved manufacturing methods, there are substantial variations in GHG emissions between structural members designed for exactly the same performance requirements. Structural designs are often not optimized because of the trade-off with constructability

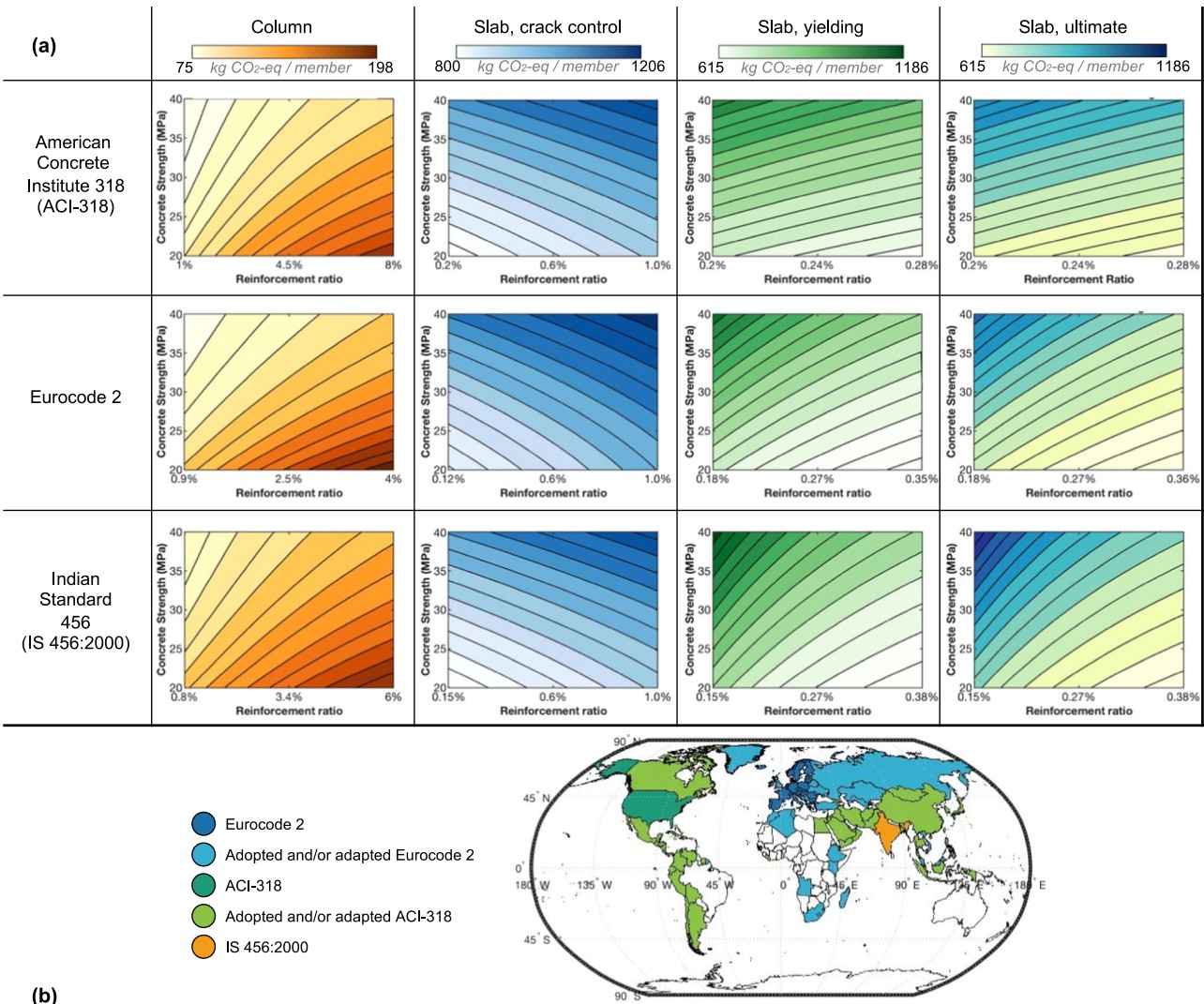

**Fig. 3 | The effects of efficient structural design on emissions from reinforced concrete members.** For the same load conditions and length/height, the compressive strength and reinforcement ratio of a member designed for compression or bending can be optimized to minimize its environmental impact. For a column, the GHG emissions are minimized for low reinforcement ratio and high compressive strength while for a beam, utilizing higher reinforcement ratio and low strength result in its lowest impact. **a** GHG emissions for example members designed following each of three design codes, considering varying concrete compressive strength, and reinforcement ratio. Note: loadings and design stages of the moment-curvature relationship vary between the column, slab designed at crack control, slab designed at rebar yielding, and slab designed at ultimate capacity; for the slab design, this work considers reinforcement ratios within allowable deflection. **b** Map of the locations where these codes, or their permutations, are being implemented (note: country borders for the map use a function written by C. Greene. "Borders". University of Texas at Austin's Institute for Geophysics (UTIG), Austin, TX. (2015)[56]). Information on code implementations and use from refs. [57–60].

efficiency, e.g., column dimensions and concrete mix designs can only vary so much for a large construction project before construction becomes over-complex, which is not economically desirable. However, these results show the potential to mitigate GHG emissions through more efficient utilization of materials in structural design, which is currently not addressed because environmental impacts are not included in the design codes. Due to the myriad members and performance requirements that exist, we limit this exploration to columns and slabs, which are among the most common reinforced concrete elements for buildings.

The effects of changing steel reinforcement ratio and concrete compressive strength to drive down materials consumption and GHG emissions were explored. Specifically, design of columns and slabs were examined as these make up a significant fraction of the built environment. Using three of the most used and adopted design codes, reflective of 105 countries, allowable member design with steel reinforcement was considered. The role of higher or lower reinforcement ratios on both GHG emissions from the amount of steel needed and the

commensurate reduction of concrete needed were examined. Simultaneously, the effects of higher concrete compressive strength on parameters such as member cross-sectional area were addressed. Here, we address emissions savings as they could accrue relative to the median reinforcement ratios and median compressive strengths considered. Details of the calculations performed and assumptions made to quantify these benefits are provided in the Methods and Supplementary Information.

Our findings show that within any accepted design code, selection of concrete strength and reinforcement ratio can result in large variations in emissions (see Fig. 3 – data in Supplementary Data 1). These variations are due to the GHG emissions per m³ of steel being much larger than per m³ of concrete, and the fact that the required cross-sectional area depends on concrete strength. As a result, by specifying different volumes of each material, while meeting design requirements such as maximum allowable slab deflection, there can be notable shifts in net GHG emissions. Notably, this work shows that low strength concrete or low reinforcement ratios do not always correlate to low

GHG emissions when the whole member design is considered, which is in line with findings by Belizario-Silva et al.[29]. For structural slabs designed at the yielding and ultimate stage, the lowest emissions for our case study loadings (discussed in the Supplementary Information) occur at the highest reinforcement ratio and lowest concrete compressive strength assessed (highest emissions occur for the lowest reinforcement ratio and highest strength). In the case when crack prevention is the controlling design factor for a reinforced concrete slab (e.g., rigid road pavement), the minimum required reinforcement ratio and low compressive strength is preferable per Fig. 3a. Dissimilar to the slab at the yielding and ultimate stage, the reinforcement does not contribute to reducing the required cross-section area of concrete and therefore, increasing the reinforcement ratio only increases the environmental impact. However, contrary to the slabs (yielding and ultimate stage), for the reinforced concrete columns, the lowest GHG emissions occur with the minimum steel reinforcement ratio and highest concrete compressive strength assessed (highest emissions occur for the maximum reinforcement ratio and lowest strength). Using an example column designed to meet the United States design code (ACI-318), the European Standard design code (Eurocode 2), and the Indian Standard design code (IS 456:2000), further differences are noted in emissions for members: a difference of >46 kg $CO_2$-eq for the column using the ACI-318 code (this is 70% greater emissions than the lowest column emissions using this code); a difference of >63.1 kg $CO_2$-eq (90% between highest and lowest) for the column using Eurocode 2; and a difference of ~51 kg $CO_2$-eq (60% between highest and lowest) for the column using the Indian Standard code. For slabs designed for bending at the ultimate stage, there is a 58–93% difference between the highest and lowest emissions members that meet design code requirements with the same boundary conditions and loading. In slab design (ultimate), there is a larger difference in GHG emissions for low reinforcement ratio than for higher ratio, which suggests that if a low ratio is used, there is increased reliance on high concrete strength or greater cross-sectional area of concrete (slab thickness), which results in higher impact. However, use of excess reinforcement is inefficient due to the significantly higher volumetric impact of the reinforcement. While trends are similar between codes used in different regions, designing slabs per Eurocode 2 and columns per ACI-318 result in the lowest impact. If all countries/regions were to design for the lowest impact per Eurocode 2 and ACI-318 for slabs and columns, respectively, it would result in a reduction of approximately 67 Gt of GHG emissions between 2015–2100 (based on a model of one unit, here defined as 1 slab + 4 columns). The authors recognize that this is a simplified model, but nevertheless useful for the argument at hand. Slabs spanning over multiple supports as well as pre- and post-tensioned slabs are common designs that could yield different results than the modeled simply supported slab. Here, it was assumed that 20% of GHG emissions are from concrete used in other applications than columns and slabs, such as in foundations. Further, if we assume a baseline of 30 MPa (the middle of the strength range considered in this work) and median longitudinal reinforcement ratio (slabs, ultimate: 0.26% reinforcement ratio and 0.45 m thickness, slabs cracking: 0.6% reinforcement ratio and 0.34 m thickness, columns: 3.5% reinforcement ratio and 0.18 m column width), then choosing the optimal combination of strength and reinforcement ratio could lower slab emissions by 20–25%, column emissions by 18–22%, and unit emissions by approximately 23% for these three codes. If instead reinforcing steel with a higher environmental impact is used, the resulting reductions are ~20% for slab, ~30% for column and ~21% for a unit (see Methods section for sensitivity analysis). However, the lower environmental impact of reinforcing steel is used in the analysis herein.

## Engineering for increased service life

This work considers that rate of population growth in urban areas is projected to be greater than overall population growth, and that urban areas are predominantly in coastal regions[3]. The use of SCMs by their incorporation in blended cements increases concrete resistance to chloride ingress, and hence durability in coastal regions[30,31]. Though carbonation is another common durability concern, specifically for corrosion of steel reinforcement, due to various factors that need to be present for corrosion to be initiated, as motivated in the Supplementary Information, carbonation was not considered in this analysis. Projections of concrete demand and associated GHG emissions were made to 2100, accounting for shifts in population growth, per capita demands of concrete in use, and national affluence (see Methods). Increased concrete durability in coastal regions was used to estimate benefits to concrete longevity (see Fig. 4—data in Supplementary Data 1). There is a twofold benefit by using a blended cement approach in this instance: (a) a substantial increase in service life, by avoiding premature deterioration due to chloride ingress (with associated repair impacts) and 'obsolescence'[32,33], and (b) that SCMs usually allow a reduction of concrete GHG emissions in comparison with plain Portland cement[34], thereby improving the sustainability of the system.

New cement production could be curtailed through improved utilization and life extension of in-stock resources for concrete systems; this curtailment in turn would influence the amount of GHG emissions associated by reducing cement production. In this work, we apply this concept by examining the material flows associated with cement and concrete production. Namely, we address regional variability in the magnitude of concrete produced, whether it is used in residential buildings, non-residential buildings, or civil infrastructure, and the current estimated longevity of the concrete in each of those applications for each of 10 regions reflecting the world. Using these baseline statistics on production, utilization, and longevity, we quantify the effects of reducing new cement demand if existing cement applications had longer service lives. Namely, by projecting future demand of concrete (accounting for differences in application, regional demand drivers such as population growth, and when existing concrete would meet limit states and require replacement), we model the effects of increasing the period before a limit state is reached. Here we assume, if a limit state is not reached, then existing concrete can stay in service and would not require new concrete to replace it. As a result, increasing this period to reaching a limit state mitigates future consumption of concrete.

Such modeling requires several assumptions regarding factors such as which regions will have access and ability to utilize SCMs for improved durability, which regions will be susceptible to deterioration and failure mechanisms that will benefit from the utilization of SCMs, and how much elongated service life can be anticipated. Here we consider a scenario in which all conventional Portland cement can be replaced with up to 50% SCMs in coastal areas with chloride-rich environments, leading to the service lifetime of new buildings being extended by threefold and other concrete systems being extended by fourfold (see estimates for service extension in Methods and the Supplementary Information). In this case, a reduction of 175.7 Gt (47.1%) GHG emissions could be achieved (see Fig. 4); even if these benefits were only achieved in half of all cases considered, this would amount to emissions reductions of ~ 25% or ~ 90 Gt, still very substantial reductions. Using other scenarios where lower levels of service life extension can be obtained or not all regions can access/utilize this high level of SCMs for improved concrete durability, or shifts in service life extension occur sooner based on current use of SCMs, we continue to see significant benefits from increasing the longevity of structures, with a range of 25–55% reduction in cement production necessary and, likewise, a 25–55% reduction in GHG emissions (see sensitivity analysis in Methods and the Supplementary Information). The authors recognize that increasing service life of concrete structures has other factors to be addressed beyond material durability, such as introduction of advancements in design offering the same functionality at lower operating costs, change in use, and change in

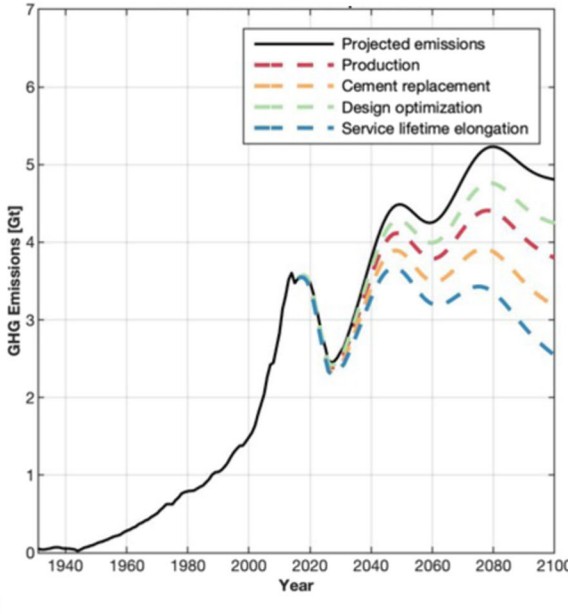

**(a)**

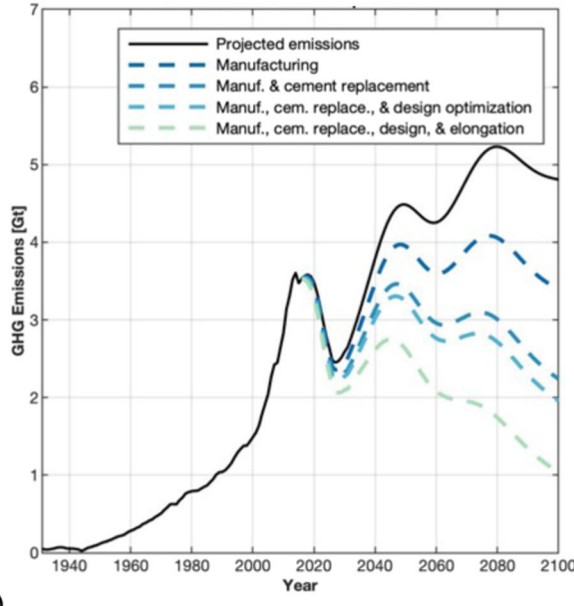

**(b)**

**Fig. 4 | Effects of manufacturing and material efficiency improvements.** By adapting mature reduction strategies, global greenhouse gas (GHG) emissions from reinforced concrete can be reduced by up to 76% in 2100 compared to a business-as-usual scenario. Reducing the demand for cement by increasing the use of SCMs and increasing structural longevity have the greatest influence. The potential reduction in GHG emissions between 2015–2100 if all methods are implemented, namely: (i) if all manufacturing interventions are considered; (ii) if the effects of increased supplementary cementitious material replacement of cement is considered; (iii) if design optimization of members is considered; and (iv) if increased structural longevity is considered. For (i)-(iii), emissions reductions are modeled here as increasing implementation linearly over time with 100% implementation by 2100; however, implementation could happen at a much faster rate. For elongating the lifetime of systems, emissions reductions are modeled as a function of stock dynamics. **a** Magnitude of emissions reduction from each measure considered separately. **b** Magnitude of emissions reduction from each measure considered cumulatively. The reduction in GHG emissions between 2018–2030, as well as subsequent drops, reflect projection models estimating annual cement production as a function of cement stock per capita and population growth. Drops in yearly emissions reflect regions such as Europe and China experiencing declining or stabilizing cement requirements as they relate to these parameters. Expected growth in population and investments in building up infrastructure in countries/regions such as India, Africa, and Developing Asia, is projected to cause a global increase in GHG emissions from cement production (note rapid increase after ~2030). (Note: Projections modeled based on data, population growth, and resource saturation predictions developed prior to the COVID-19 pandemic; as new data are accumulated, future work should account for the effects of this pandemic on concrete demand).

standards or legislations that impact the structure's economic service life. In addition, for structures to be in use longer, retrofit and maintenance might be necessary, resulting in additional material consumption. Despite these limitations, it is critical that we consider the service life of our systems as substantial environmental benefits can be achieved by leveraging shifts in material use at this stage.

Projecting cement demand and emissions from production, the cumulative effects of the strategies previously discussed are examined between 2015–2100: all manufacturing improvements would lead to a 21% reduction in GHG emissions; increased use of fly ash and slag (GGBS) as SCMs would lead to 11% (at 30% replacement) to 34% (at 50% replacement) reductions in GHG emissions; optimizing concrete strength and steel reinforcement for building applications would lead to 18.5% reduction in GHG emissions; increasing concrete system longevity by up to fourfold from current regional average service lives, resulting in a world cement demand reduction by 47.1% and 175.7 Gt of $CO_2$-eq emissions. Depending on the global population growth, the resulting global cement demand will vary, and hence the generated GHG emissions. A sensitivity analysis for a low and high population growth scenario can be found in Supplementary Information.

## Discussion
The degree of GHG emissions mitigation possible through the efficient use of cement and concrete, achieved through design improvements within current design codes, could be used to inform engineers, material scientists, policy makers (or code-writers), and other stakeholders. Many regulatory bodies have emphasized the need for all $CO_2$ emissions to achieve net zero within the coming decades in accordance with the findings of the Intergovernmental Panel on Climate Change[6]. A focus on cement production related emissions has been noted in attempts to meet goals set out in the Paris Agreement[35] and in regional regulations (e.g., California's Bill to eliminate GHG emissions from cement[36]). However, due to the difficulties in fully eliminating limestone decarbonation emissions from cement production[7], material efficiency strategies that limit the demand for these materials will be a critical aspect of meeting emissions goals[1,18]. This needed mechanism for reducing emissions has been accepted by industry as well (e.g., the Global Cement and Concrete Association[27]). Here, we show that within already accepted design, there is huge opportunity to reduce emissions. The inclusion of environmental impact assessments to calculate GHG emissions reduction within conventional codes, material specifications, and procurement/design decisions is critical. Because such methods can already be implemented, they should be put into effect immediately.

## Methods
In this work, we compile the anticipated global GHG emissions savings from using several key engineering strategies to reduce emissions during concrete production and efficiently use cement-based materials. To do this, we estimate the GHG emissions to produce cement-based materials worldwide (i.e., cradle-to-gate emissions). We then adapt these models to reflect the influence that a variety of manufacturing improvements would have on reducing GHG emissions. Finally, we study the influence that design-based improvements (namely, concrete mixture proportioning, selecting appropriate steel reinforcement ratios to meet design standards while limiting

emissions, and improving concrete in-service use periods) can have on reaching net-zero GHG emissions goals. The methods for this work are presented below, and details of the methodologies and data used are presented in the Supplementary Information.

## Emissions baseline

Emissions from the production of cement-based materials were modeled for six representative years: 1990, 1995, 2000, 2005, 2010, 2015. For each of these years, kiln efficiency, thermal energy mix, electricity demand, and SCM content for the cementitious materials (i.e., Portland cement and SCMs) were determined based on the Getting the Numbers Right (GNR) Initiative[37,38]. The electricity mix for the full production system was based on world average for the same years from the International Energy Agency (IEA)[39]. Remaining concrete constituents, including aggregates and water, were determined as a function of cement use, namely in either concrete or mortar (all non-concrete uses of cement were modeled as containing constituents equivalent to mortar); calculation methods for these ratios are presented in the Supplementary Information. Concrete demand was broken down by strength class, and distributions of concrete constituents were determined using a database of concrete mixtures collected from the academic literature and representative of common concrete constituents and strengths used globally (see discussion of data in Supplementary Information). Distributions were fitted to concrete mixtures in this dataset that fell within the European Ready Mixed Concrete Organization (ERMCO)-specified strength groupings; in these distributions, all cementitious materials were grouped together to form a distribution, which was then sub-divided based on the mineral additive content reported by the GNR Initiative[37]. To estimate mortar constituents, distributions were modeled based on the standard mortar constituents reported by ASTM International[40]. As noted, we model non-concrete, cement-based materials as having approximate constituents of mortar. We note this simplification does not capture all cement-based materials. However, due to poor data availability for other cement-based products, and the largest fraction of non-concrete products being mortar, this is an accepted approximation in the academic literature[41].

GHG emissions from the production of concrete and mortar in 2015 (on which date all mitigation results are based, and modeled based on each kg of cement used) was considered to be the baseline herein. These emissions were projected forward to 2100 by using the same emissions per kg of cement consumed and the quantity of cement required in future years, i.e., a do-nothing or business-as-usual scenario. The authors note that the data prior to 1990 were limited; as such, the emissions per kg of cement for all years prior to 1990 were assumed to be equivalent to emissions per kg of cement in 1990, and changes in cumulative emissions in those years were reflective of differences in the quantity of cement produced annually.

## Projection of future cement demand

To estimate future cement demand, requirements for 10 countries and regions were projected using the model developed by Cao et al.[24]. In this modeling approach, the use of cement was broken down by category: residential buildings (Res), non-residential buildings (NonRes), and other civil engineering applications (CE). Because the model by Cao et al.[24]. focused on production after 1950, for this work, historic data for cement inflow is used for the years 1931–1950. Then, the cement inflow between 1951 and 2100 was captured based on a stock-driven approach, following the procedure outlined by Cao et al.[24], which uses per capita saturation levels (i.e., the upper threshold of per capita demand per person per year), the period of time cement-based materials stay in-use (i.e., the longevity of cement-based materials in-stock), and population projection statistics (based on United Nations data) synthesized by Cao et al.[24]. Projected cement consumption was determined through use of a combined Gompertz model which calculates the growth curve of

future per capita material stock based on in-service lifetimes and stock patterns[24]. The service lifetimes ranged from ~31 to 100 years for Res, ~31 to 76 years for NonRes, and ~30 to 75 years for CE applications. Generally, shorter in-use lifetimes are seen in developing countries/regions for all three applications. The in-use service life was modeled as a non-deterministic value, i.e., all cement is not taken out of service when the average service life for the country/region in particular is reached, and therefore the cement staying in use is modeled as a distribution, again based on by Cao et al.[24].

To capture anticipated changes in populations, projected population data from the UN World Population Prospects[42] were applied using the medium variant. A sensitivity analysis was performed to capture the impact of alternative population growth patterns between 2020–2100, using the low and high population variant, also from the UN World Population Prospects[42]. Results from the sensitivity analysis are presented in the Supplementary Information. The ten countries/regions modeled were: (1) North America; (2) Latin America; (3) Europe; (4) Commonwealth of Independent States (CIS); (5) China; (6) India; (7) Africa; (8) the Middle East; (9) Developed Asia & Oceania; and (10) Developing Asia. Cement demand was calculated based on inflow for each application (namely, Res, NonRes and CE) for each of the countries/regions noted.

## Concrete manufacture and mixture proportions

To examine the effects of typical differences in GHG emissions as a function of concrete constituents and manufacturing methods, the same concrete mixture dataset was used. Mixtures within ±3 MPa of 20 MPa, 35 MPa and 50 MPa compressive strength at 28 days were used for comparisons; mixtures had varying water-to-binder ratios, SCM replacement levels, and aggregate contents to achieve the same 28-day strength.

Calculations of GHG emissions to produce concrete mixtures were performed for cradle-to-gate production. That is, these calculations included sources of GHG emissions from raw material acquisition through constituent mixing, but not including placement or other construction-related emissions. For this work, it is assumed that changes proposed will have limited effect on construction, use phase, or end-of-life GHG emissions of concrete between alternatives; therefore, GHG fluxes that occur in life cycle phases subsequent to concrete production are modeled as equivalent between alternatives and not incorporated into calculations. The emissions for each mixture were calculated as a volumetric impact, i.e., in terms of kg $CO_2$-eq emissions per $m^3$ of concrete (kg $CO_2$-eq / $m^3$). The GHGs considered in this work are $CO_2$, $CH_4$, and $N_2O$, and they were assessed in terms of $CO_2$-eq using the 100a global warming potentials from the Intergovernmental Panel on Climate Change (IPCC)[43]. Details on how environmental impact assessments were performed and further assumptions made are presented in the Supplementary Information.

To assess changes that could alter GHG emissions during cradle-to-gate production of concrete, this work examines alterations in both cement manufacturing methods and concrete mixture proportions (focusing on increased utilization of SCMs). For manufacturing methods, this work focuses on the beneficial effects of commonly discussed GHG emissions mitigation strategies during cement production, namely: increasing kiln efficiency, switching higher emitting kiln fuels to natural gas, switching higher emitting electricity resources to wind power, and a combination of these strategies.

Further, to address utilization of SCMs, initial assessment included impacts associated with inclusion of Limestone filler (LS), Natural Pozzolans (NP), Shale Ash (SA), Calcined Clay (CC), Silica Fume (SF), Fly Ash (FA) and Blast Furnace Slag (GGBS) in concrete mixtures. Ranges in environmental impacts from the use of increased SCM content were based on an increase from the 2015 average SCM content (20.3%), as reported by the GNR Initiative[37], to 30% and 50% SCM content[44] The authors note that the supply of GGBS and FA may decrease in the future,

but we anticipate that similar performance can be achieved by utilization of NP, and though the availability of certain NP is regional, a wide range of pozzolanic materials could be used (e.g., tuff, calcined clays, agricultural byproducts)[45,46]. Due to variations in GHG emissions from production of different SCMs (see Supplementary Information), additional SCM content is modeled here as having equivalent emissions to NP.

These more commonly discussed strategies can be compared to mitigation possible from structural design to provide context for the significance of design for efficient concrete use as a mitigation method. Further, these alterations were used to assess a cumulative potential mitigation of GHG emissions from global concrete production and use.

### Design and application

This phase of study examines the influence of the amount of reinforcing steel and concrete compressive strength in design of concrete members in buildings, to ascertain their ability to contribute to mitigation of GHG emissions. To determine the effect of these factors, this work couples two models from the literature: (1) the mixture proportioning relationships developed by Fan[25] to link GHG emissions from concrete production to its compressive strength and (2) the methods developed by Kourehpaz[26] to link concrete compressive strength and reinforcement ratio to environmental impacts. To apply the relationships derived by Fan[25], parameters were derived for the effects of binder content as it relates to both GHG emissions and concrete compressive strength (derivation and parameters determined are presented in the Supplementary Information). The method developed by Kourehpaz[26] was based on reinforced concrete members meeting the ACI 318 design code from the American Concrete Institute[47]. In this work, similar relationships were derived to examine design of reinforced columns (3.5 m standard height) and reinforced slabs (spanning 7 m) at cracking, yielding of reinforcing steel, and ultimate stages according to equivalent members based on Eurocode 2[48] and the Indian Standard code[49]. The slabs are designed for bending, and while other structural design aspects such as shear can affect the GHG emissions, those are outside the scope of this analysis. The equations were developed to allow the cross-section area of the column to vary for a fixed applied axial load, and the slab depth to vary for a constant uniform load, for a range of compressive strengths and reinforcement ratios. More detail on assumptions made and application are presented in the Supplementary Information. To address the GHG emissions for reinforcing steel, this work assumes the steel used contains 80% recycled content, with an impact of 1.03 kg $CO_2$-eq per kg of steel (based on[50], which accounts for recycled content of steel); however, the authors note that different production methods for steel (e.g., using a blast furnace or electric arc furnace, different recycled content, and different energy mixes in manufacturing) can lead to different GHG emissions. To address variations in manufacturing methods, a sensitivity analysis was performed to examine the impact of using reinforcing steel with higher environmental impact. The higher value used, 2.29 kg $CO_2$-eq per kg of steel, was chosen based on ref. 50. Contribution from reinforcement other than longitudinal steel, such as stirrups, mesh, or shear reinforcement, have not been considered in this study for simplicity.

### Increased service life

To analyze the effects on GHG emissions of service life extension for concrete structures and mortar used, several factors were assessed concurrently. This work uses the model developed by Cao et al.[24]. for estimating the in-use cement stock, and scales this stock to estimate cement-based materials demand based on the models discussed in depth in the Supplementary Information. Again, this model categorizes cement demand in three different applications (Buildings (which include Res and NonRes), and CE). The ratios of use in each of these sectors and the longevity of service were based on data collected by Cao et al.[24], where the service-life modeling is based on a statistical analysis of longevity of concrete systems in the herein 10 countries/regions, that cumulatively represent the world.

The role of mixture selection and structural design on the longevity of concrete systems was analyzed by altering the in-stock time-horizons modeled for each concrete application. A further benefit is that concretes with blended cements generally have much higher resistivity than plain Portland concretes, which allows possible steel corrosion to be better controlled and managed during the active corrosion phase, thus further extending the service life[30,31,51]. Binder selection is critical in extending concrete service life. Taking chlorides as the illustrative case for durability: the expected corrosion-free service life of a concrete structure with 40 mm steel cover in a typical marine environment, using different binder types (plain Portland cement; Portland cement with blends of either 30% FA, or 50% GGBS) can be estimated, following[30,31]. Based on the generally accepted critical chloride threshold of 0.4% chlorides by mass of binder at the steel as the corrosion initiation threshold, different corrosion-free life can be expected for different binder systems: <5 y for OPC, ~25 y for FA, and ~50 y for GGBS. The problem here is simplified, but nevertheless illustrates a possibility of extending service life substantially by judicious choice of concrete materials, while at the same time retaining all necessary mechanical and physical properties of the concrete. While using SCMs may increase the ability of a concrete structure to carbonate[52], this does not necessarily translate to greater corrosion-susceptibility, as indicated in the Supplementary Information.

This work considers that the mean service lifetime for each of the three application categories was increased, as a result of improved durability, by threefold and fourfold (see Supplementary information) for buildings and infrastructure, respectively, with various service-life extension scenarios considered based on cement use. To estimate the potential effect cement replacement and longer service lives could have had retrospectively, an ideal scenario was modeled, where it was assumed that all countries/regions would have access to 50% SCM and as a result, service lives can increase by fourfold. A more realistic scenario was modeled as well, where it was assumed that only certain parts of the world (50%) will have access to SCMs to replace up to 50% of cement in concrete. For this scenario it was also assumed that access to SCMs would have increased since 1931, and hence a fourfold and a threefold service life extension was modeled for historical and projected data. Results from the sensitivity analysis based on these scenarios are summarized in the Supplementary Information. The service life was modeled as a distribution, and hence all cement in-use was not taken out of service at the same time to reflect the effective in-use time of buildings and infrastructure. This life extension was based on empirical findings on the influence of using FA and GGBS on hindering corrosion of steel reinforcement in coastal regions, and the associated increase in service life of the concrete structures[30,31]. It was assumed that this level of increased longevity would be attained with a 30 or 50% SCM content. The GHG emissions from producing these SCMs was modeled here as having equivalent GHG emissions to natural pozzolans. Due to higher levels of SCMs being utilized within recent years in cement and concrete production, namely a ~10% rise in use has been reported within the past 30 years[53], prolonged use was considered to begin with structures currently in-stock for one of the idealized scenarios; however, for the baseline considered in the manuscript, only future stock (2015–2100) is considered to have elongated service life. It was further assumed that the increase in service life would contribute to efficient utilization of resources and, thus, decrease the demand for new cement.

### Cumulative effects of strategies considered

At this stage of assessment, the mitigation potential from each strategy was considered. In this component of the analysis, the mitigation potential from using improvements in manufacturing (e.g., increased kiln efficiency), lower levels of clinker per cubic meter of concrete achieved through use of SCMs, desired concrete compressive strength

and reinforcement ratio for infrastructure systems, and increasing longevity from SCM use were examined cumulatively. To assess the influence of compressive strength and reinforcement ratio on GHG emissions at a global scale, differences in emissions were calculated based on strengths, SCM content, and manufacturing methods that would result in lower GHG emissions relative to the global average in the year 2015, and reinforcement ratios that would result in lower GHG emissions relative to the mean of the reinforcement ratios.

To scale member designs to global infrastructure systems, two primary approaches were taken. Concrete used in civil infrastructure was not considered herein for improvements to reinforced members as such concrete can vary from reinforced to substantial uses as non-reinforced or nominally reinforced concrete (e.g. dams, pavements, mass concrete applications etc.); therefore, we model these applications as not to be reinforced like buildings[54]. For buildings, the member designs were extended to reflect concrete structures based on a method proposed by Schmidt et al.[55] namely, the requirements of members in terms of strength and cross-sectional area were related to relative volume of horizontal members (e.g., flat slabs) and vertical members (e.g., columns) as well as the degree of loading related to the height of the structure. A simplified approach was applied herein where both NonRes and Res buildings were modeled as having an average height of 3.5 m per story. Vertical members were modeled based on the reinforced column design methodology, and horizontal members were modeled based on the slab designed for bending at the ultimate stage methodology. This method allowed for the assessment of mitigation potential from median reinforcement ratios and from the current average strength concrete used around the world. More detail on assumptions made and application are presented in the Supplementary Information. Code to reproduce the figures in the manuscript have been made available by the authors.

## Data availability
The concrete mixture data, the design data, and the consumption data used in this study can be found in the Supplementary Information.

## Code availability
Code to reproduce the figures in the manuscript is available at https://doi.org/10.25338/B8793W.

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

## Acknowledgements

S.M. acknowledges funding provided by the National Center for Sustainable Transportation and the California Department of Transportation (65A0686/TO 027, S.M.) and the National Science Foundation (CBET 2143981, S.M.). S.M. and J.O. acknowledge funding in the form of a gift from the ClimateWorks Foundation (gift funds, S.M.). This work represents the views of the authors, not necessarily those of the funders.

## Author contributions

S.M. developed the methodological approach. J.O. collected the data and evaluated the results. J.O., M.A., and S.M. wrote and edited the manuscript.

## Competing interests

The authors declare no competing interests.
