## [Peer Review File · Nature Communications]

Near-term pathways for decarbonizing global concrete productionREVIEWER COMMENTS

Reviewer #1 (Remarks to the Author):

This manuscript presents some potentially very interesting thoughts and results. It advances beyond previous arguments and adds value particularly in the comparison between different building codes, and in quantifying potential savings that may be achieved through different decarbonisation strategies, in a way that I haven't seen done before.

However, I cannot recommend its publication. This is fundamentally founded in the fact that the structure of the paper doesn't work as it is currently written. It is evident that the authors have worked hard to fit the manuscript into the length and format limitations of Nature Communications, but unfortunately this makes it nearly unreadable.

The Supporting Information file contains almost everything of real technical value to a specialist in the field, while the main text is essentially an extended "discussion" section, and the fact that the Methods section comes after this means that the paper more or less has to be read backwards, because the preceding discussion really can't be properly understood before the Methods section has been read and comprehended. The Figure captions then have an inordinate amount of information shoe-horned into them as well; this makes everything just a little too convoluted for clarity of reading.

I can see what the authors are trying achieve - and I understand that it's a constraint imposed by the Communications format of the journal - but this manuscript just doesn't work in this style. If it was put back together into a more conventional structure, with the Methods at the start and the SI file re-integrated into the main text, it would be a commendably interesting paper for any number of journals as a full-length research article, and my encouragement to the authors is that this would probably be the best way to gain value from this important work.

Reviewer #2 (Remarks to the Author):

The paper is interesting because compares the effect of the main codes for structural design on the GHG emissions of concrete structures. It is a first attempt to assess the effect of the structural codes and decisions made by the structural designers. There are many limitations related to the assumptions made by authors. In my opinion these limitations should be clearly stated and emphasized. The quality of data seems good but in order to check calculations it would be useful if authors could make available the database. In different points references needs to be improved.
Below some suggestions.

Main Paper

Page 4 line 2. GCCA

Page 5 line 18-19. Please check the sentence, it is not so clear.

Page 5 line 21-22. Other works already showed this concept (<https://doi.org/10.1016/j.cemconcomp.2010.07.009> and <https://doi.org/10.1016/j.job.2021.102979>)

Page 6 line 25 and pag 7 line 1. "(based on a 1 model of "one unit" (1 slab + 4 columns)". This is the limitation of the model of this work, I think it should be emphasized.

Page 7 line 6. When you talk about "reinforcement ratio" I assume you are looking for the longitudinal reinforcement. I would like to know if the contribution of stirrups (in case of columns and beams) and mesh (in case of slabs) has been taken into account. If not, you should declare this point.

Page 7 line 12-14. Please support with references the sentence about the performances of blended cements against chloride ingress. What about carbonation? You can complement the sentence and add some references.

Page 7 line 19. Ref 24,25. You could improve the references.

Page 7 line 22-23. Many times, structural retrofitting is necessary. The authors could mention this point.

Page 8 line 21. What type of SCMs you are talking about?

Page 8 line 24. How many years you are thinking about? Usually for building SL is 50years up to 100years for infrastructures.

Page 10 line 22-23. Please check. Standard mortar constituents are quite different from market products.

Page 10 line 25. Which model has been used to forecast up to 2100? What kind of considerations have been made to take into account the typical cyclicity of the construction market?

Page 11 line 2. The "do nothing scenario" is the business-as-usual?

Page 11 line 4-5. Please comment how the assumption affects the results.

Page 13 line 10. Authors note that different production methods can lead to different GHG emissions. I suppose they did not assess the different production methods. Probably they should and propose a range.

Page 14 line 5. Concretes with blended cement have much higher resistivity. Resistivity depends on the saturation degree of the material. Please support your sentence with references.

Page 14 line 9. 3-fold and 4-fold. I suggest writing even the approximate number of years.

Page 14 line 10-12. "This life extension was based on empirical findings on the influence of using FA and GGBS on hindering corrosion of steel reinforcement in coastal regions, and the associated increase in service-life of the concrete structures." Please support the sentence with references.

Supplementary info

Page S3. "Even though supplies of these two well-established industrial byproducts may decrease in the future we anticipate that other mineral additives (e.g., natural pozzolans) will contribute to similar performance." This sentence is strong because the availability of natural SCMs is regional. I suggest reformulating the sentence.

Page S5 end of the paragraph. I think this database should be increased. 3 out of 5 references come from the same authors and literature is rich of works containing mix proportion data.

Page S7. Although in the EN1991-1-1 Annex A, the unit weight of steel γ is specified between 7700 kg/m³ and 7850 kg/m³ is common in structural analysis the last value. I think this does not affect the results, but I suggest using 7850kg/m³

Page S7 equation (5). Please specify in the text what kind of approximation the equation introduces?

Page S13. Authors consider $E_c = 17\text{GPa}$. Why do you not use the relationship used in the codes to calculate E_c as function of F_{ck} ? Above you consider F_c between 20 and 40 MPa.

Page S18. "for every kg of cement used, there are on average 7.3 kg of cement-based materials". This means 325kg of cement per cubic meter of concrete. Right? So you have the total cement production/consumption, you divide this amount per 325kg/m³ and determine the volume of concrete produced by that amount of cement. If my thinking is correct, you should rewrite the equation to improve the readability. If I did not understand, please clarify because is a little bit confusing. Verify even if eq 59 needs to be rewritten.

Table S5. "Mean" is it the mean of the median values?

Page S25 item S.6.6. This analysis is preliminary. As authors claim, many aspects related to structural modeling have not been considered. Besides foundations, roofs etc., depending on the height of the building other structural elements are necessary. So, I think that the note at the second paragraph should be moved to the 1st paragraph and clearly state the limitation.

Reviewer #3 (Remarks to the Author):

The authors present a detailed study of mitigation strategies not using still prospective technologies. Their paper is interesting, useful and well laid out. There are however some issues which have not been addressed which would help. Notably sensitivity studies are required for some parameters.

The introductory paragraph sets the scene but it would be better to give a sense of the percentages,

rather than "1 billion" which may or may not be a lot of concrete in relative terms. In particular, for their projections, the authors should consider a range of scenarios, in terms of population growth and urbanisation.

The sentence "Critically, the efficient use of these materials must be a step in mitigating GHG emissions from material production" is cryptic. Do the authors mean that geopolymers cannot ever abate emissions because the use resources that are used 1:1 in abating cement use and adds activators? They should say so clearly.

p 3 There are also alternative structural systems which have been proposed and which could be more efficient. In general, missing from the study is the impact of e.g. using more steel frames instead of concrete frames where possible.

p 5 The authors should compare tailoring mixes to requirements with the number of mixes already used on-site. Are they proposing to make construction much more complex?

p 6 There can be other considerations, in particular acoustic comfort which may impose a minimum depth for the slab. Have the authors looked at the scope for optimisation assuming a sensible range of slab depths?

p 7 are the authors taking into consideration punching shear design? thicker slabs may require less shear reinforcement...

p 8 the authors need to more critically look at the reasons for demolitions. It is very uncommon that buildings be demolished due to reaching the end of their service life. much more common that they reached the end of their economic lives. The authors should do a sensitivity analysis looking at the impact of their proposed measure as a function of the effective lives of buildings (40-80 years)

p 12 The authors should also consider the variation in cement amount in mix designs in "real life", to add error bars to their analysis...

Title: Near-term pathways for decarbonizing global concrete production
Authors: Josefine Olsson, Sabbie A. Miller, Mark G. Alexander
Manuscript Number: NCOMMS-23-03837-T

REVIEWER COMMENTS

Note: Page and line numbers in our responses refer to the revised manuscript and SI.

Reviewer #1:

Comment R1.1: This manuscript presents some potentially very interesting thoughts and results. It advances beyond previous arguments and adds value particularly in the comparison between different building codes, and in quantifying potential savings that may be achieved through different decarbonisation strategies, in a way that I haven't seen done before.

However, I cannot recommend its publication. This is fundamentally founded in the fact that the structure of the paper doesn't work as it is currently written. It is evident that the authors have worked hard to fit the manuscript into the length and format limitations of Nature Communications, but unfortunately this makes it nearly unreadable.

The Supporting Information file contains almost everything of real technical value to a specialist in the field, while the main text is essentially an extended "discussion" section, and the fact that the Methods section comes after this means that the paper more or less has to be read backwards, because the preceding discussion really can't be properly understood before the Methods section has been read and comprehended. The Figure captions then have an inordinate amount of information shoe-horned into them as well; this makes everything just a little too convoluted for clarity of reading.

I can see what the authors are trying achieve - and I understand that it's a constraint imposed by the Communications format of the journal - but this manuscript just doesn't work in this style. If it was put back together into a more conventional structure, with the Methods at the start and the SI file re-integrated into the main text, it would be a commendably interesting paper for any number of journals as a full-length research article, and my encouragement to the authors is that this would probably be the best way to gain value from this important work.

Response R1.1: The authors appreciate the reviewer taking the time to provide thoughtful feedback on this paper. To address the reviewer's concern regarding the format of this paper, a summary of the methods used in this study was added to the introduction of the manuscript. We have also provided more context in the manuscript regarding the performed analyses and comparisons and moved content from the Supplementary Information to the Methods section to provide more clarity of the work.

Regarding the figure captions, we have provided summary statements at the initiation of each caption to highlight main takeaways. We then use the remainder of the caption to provide the more detailed explanations of concepts. Page 4, page 5 (line 6-14), page 7 (line 7-17), page 10 (line 10-21), page 14-15, page 17 (line 22-25), page 19-20

Reviewer #2:

Comment R2.1: The paper is interesting because compares the effect of the main codes for structural design on the GHG emissions of concrete structures. It is a first attempt to assess the effect of the structural codes and decisions made by the structural designers. There are many limitations related to the assumptions made by authors. In my opinion these limitations should be clearly stated and emphasized. The quality of data seems good but in order to check calculations it would be useful if authors could make available the database. In different points references needs to be improved.
Below some suggestions.

Response R2.1: The authors would like to thank the reviewer for taking the time to provide productive and detailed feedback on this paper. All data used in this work are present in the supporting information and in the references cited within the Supporting Information. However, to improve data clarity, we have added an additional Supporting Information excel file that contains the data in Figures 2, 3, 4, and S1.

Main Paper

Comment R2.2: Page 4 line 2. GCCA

Response R2.2: We have corrected the noted typographical error. (Page 5, line 15)

Comment R2.3: Page 5 line 18-19. Please check the sentence, it is not so clear.

Response R2.3: The authors appreciate the reviewer's critical and thoughtful assessment and have rephrased the line highlighted to clarify that large variations in results are caused by difference in GHG emissions between steel and concrete. (Page 7, line 19-21)

Comment R2.4: Page 5 line 21-22. Other works already showed this concept (<https://doi.org/10.1016/j.cemconcomp.2010.07.009> and <https://doi.org/10.1016/j.job.2021.102979>)

Response R2.4: The authors appreciate the reviewer taking the time to list additional work on the impact of reinforcement ratio and strength on the GHG emissions. To the best of the authors knowledge no other work has studied the potential reductions from combining optimization of mixture design with optimized reinforcement and concrete (strength and volume) utilization, particularly in combination with other GHG emissions mitigation strategies considered. However, the listed references have been added as references to clarify that similar work has been done. (Page 6, line 6 and page 8, line 1).

Comment R2.5: Page 6 line 25 and pag 7 line 1. “(based on a 1 model of “one unit” (1 slab + 4 columns)”. This is the limitation of the model of this work, I think it should be emphasized.

Response R2.5: The authors agree with this point and have added a clarification after the line highlighted to clarify that this is a limitation of the model. (Page 9, line 4-6)

Comment R2.6: Page 7 line 6. When you talk about “reinforcement ratio” I assume you are looking for the longitudinal reinforcement. I would like to know if the contribution of stirrups (in

case of columns and beams) and mesh (in case of slabs) has been taken into account. If not, you should declare this point.

Response R2.6: The authors agree with this point and have rephrased the line highlighted to clarify that the reinforcement ratio refers to longitudinal reinforcement only. (Page 9, line 9)

Comment R2.7: Page 7 line 12-14. Please support with references the sentence about the performances of blended cements against chloride ingress. What about carbonation? You can complement the sentence and add some references.

Response R2.7: The authors appreciate the reviewer making this critical point regarding the performance of blended cements against chloride ingress. Supporting references have been added, and the authors would also like to clarify that the reason for only considering chloride ingress in this analysis is motivated in the Supplementary Information. However, we have added a clarification to the reasoning behind using chloride ingress only to the section about increased service life. (Page 9, line 22-25)

Comment R2.8: Page 7 line 19. Ref 24,25. You could improve the references.

Response R2.8: The authors have added new citations to better substantiate the point being made in the line highlighted by the reviewer. (Page 10, line 5)

Comment R2.9: Page 7 line 22-23. Many times, structural retrofitting is necessary. The authors could mention this point.

Response R2.9: To address this point, the authors have added a sentence recognizing that retrofitting might be necessary for structures to be in use longer on page 11 (line 15-16) and that this might need to additional material consumption.

Comment R2.10: Page 8 line 21. What type of SCMs you are talking about?

Response R2.10: The authors appreciate the reviewer pointing out that specific SCMs are not mentioned in this section. We have clarified that the SCMs referred to are fly ash and slag. (Page 11, line 21)

Comment R2.11: Page 8 line 24. How many years you are thinking about? Usually for building SL is 50years up to 100years for infrastructures.

Response R2.11: The authors appreciate the reviewer's note regarding years in-service. In this model, we are using average service life for buildings (residential and non-residential separated) and infrastructure in 10 global regions, varying from ~30-75 years for infrastructure and ~30-100 years for buildings, based on the model by Cao et al. (cited in SI and Methods section). In addition to clarification regarding the service-lives used added to the SI, to improve clarity of the manuscript, we have added a section to the Methods explaining the service-life modeling in general. Further clarification to the used model, which is based on statistical analysis of

longevity of concrete systems in 10 regions that cumulatively represent the world, has been added in the manuscript. (Page 11 (line 24-25), page 14-15, and page 18 (line 20-22))

Comment R2.12: Page 10 line 22-23. Please check. Standard mortar constituents are quite different from market products.

Response R2.12: The authors agree that there can be a wide range of constituents used in non-concrete cement-based products in the market. However, data availability for these constituents are quite poor. In order to perform a preliminary estimate of these constituents, we approximate non-concrete mixtures as being mortar, which is the dominant non-concrete product worldwide. This assumption is grounded in the academic literature as a means to approximate these non-concrete, cement-based products. We have clarified this point in the text and cited an appropriate reference justifying this approximation. (Page 14, line 1-5)

Comment R2.13: Page 10 line 25. Which model has been used to forecast up to 2100? What kind of considerations have been made to take into account the typical cyclicality of the construction market?

Response R2.13: The authors appreciate the reviewer bringing up this question. We have added a new section to the Methods, describing the modeling assumptions and inputs for the projection (previous Section S.6.1 in the SI). Regarding the typical cyclicality of the construction market, we are using a material flow model based on global population projection, cement in-use per capita (projected) and average service life for buildings and infrastructure in the world regions considered. (Page 14-15, and SI page S20)

Comment R2.14: Page 11 line 2. The “do nothing scenario” is the business-as-usual?

Response R2.14: The authors have clarified that the “do nothing” scenario is the business-as-usual scenario. (Page 14, line 9-10)

Comment R2.15: Page 11 line 4-5. Please comment how the assumption affects the results.

Response R2.15: The authors thank the reviewer for this comment. The assumption for the data prior to 2015 does not impact the results as reductions and results are based on “future” data between 2015-2100. We have clarified in the text that all emissions mitigation results are based on production after 2015. (Page 14, line 6-7)

Comment R2.16: Page 13 line 10. Authors note that different production methods can lead to different GHG emissions. I suppose they did not assess the different production methods. Probably they should and propose a range.

Response R2.16: The authors appreciate the reviewer noting the likely range in emissions from steel reinforcement production. To address this point, we have run the model for a high impact steel in addition to the steel with high recycled content. The emissions reductions for design of slabs and columns, as well as the unit, decreased as a result. We present this range in the manuscript. See page 9, line 13-16 and page 18, line 5-8.

Comment R2.17: Page 14 line 5. Concretes with blended cement have much higher resistivity. Resistivity depends on the saturation degree of the material. Please support your sentence with references.

Response R2.17: The authors have added references to support this statement. (Page 19, line 7)

Comment R2.18: Page 14 line 9. 3-fold and 4-fold. I suggest writing even the approximate number of years.

Response R2.18: The authors thank the reviewer for making this point. In this model, we are using average service life for buildings and infrastructure for 10 world regions, with great variability, and therefore there is no single number of years of service life increase. To address this suggestion, we have added service-life ranges for the three different applications to the Methods section and Supplementary Information, and as noted in response to Comment R2.11, we have improved the clarity of how service life was modeled in the manuscript. (Page 11 (line 24-25), page 14-15, page 18 (line 20-22) and SI, page S20)

Comment R2.19: Page 14 line 10-12. “This life extension was based on empirical findings on the influence of using FA and GGBS on hindering corrosion of steel reinforcement in coastal regions, and the associated increase in service-life of the concrete structures.” Please support the sentence with references.

Response R2.19: The authors appreciate the reviewer’s note regarding the need for supporting references. References have been added to this statement regarding the influence of using fly ash and ground granulated blast furnace slag on service life of reinforced concrete structures in coastal regions. (Page 20, line 8)

Supplementary info

Comment R2.20: Page S3. “Even though supplies of these two well-established industrial byproducts may decrease in the future we anticipate that other mineral additives (e.g., natural pozzolans) will contribute to similar performance.” This sentence is strong because the availability of natural SCMs is regional. I suggest reformulating the sentence.

Response R2.20: The authors appreciate the reviewer making this point. We have added a statement to acknowledge that the availability of certain natural pozzolans is regional, but that there are a wide range of pozzolanic materials that could be used (e.g., tuff, calcined clays, agricultural byproducts). (Page 16, line 20-24 and SI page S3)

Comment R2.21: Page S5 end of the paragraph. I think this database should be increased. 3 out of 5 references come from the same authors and literature is rich of works containing mix proportion data.

Response R2.21: The authors would like to thank the reviewer for bringing up this point. We have intentionally selected this limited number of sources due to our goal to maintain consistency in mixture design, including type of cement, how mineral additives were integrated

into the paste, and consistent degrees of water use. We note there are many additional permutations one can make of concrete mixtures; this consistency facilitates robust modeling. However, others can extend our modeling efforts to additional mixtures if they desire to build from this work. We have included our reasoning for the selection of these mixtures in the Supporting Information. (Page S5)

Comment R2.22: Page S7. Although in the EN1991-1-1 Annex A, the unit weight of steel γ is specified between 7700 kg/m³ and 7850 kg/m³ is common in structural analysis the last value. I think this does not affect the results, but I suggest using 7850kg/m³

Response R2.22: The authors appreciate this suggestion, and the reviewer is in fact correct that this minor difference would not change the results. We ran an initial simulation and showed that a switch to 7850 kg/m³ would result in <1% difference in results. We have added to our Supplementary Information a note to this effect. Namely: "The authors note there is a minor range in densities possible for steel reinforcement, and our models suggest less than 1% difference in GHG emissions findings would occur for the column and slab modeled if density of reinforcing steel were to increase by 50 kg/m³" (Page S7)

Comment R2.23: Page S7 equation (5). Please specify in the text what kind of approximation the equation introduces?

Response R2.23: The authors have added a clarification to equation (5) in the text below the equation noted. (Page S7)

Comment R2.24: Page S13. Authors consider $E_c = 17\text{GPa}$. Why do you not use the relationship used in the codes to calculate E_c as function of F_{ck} ? Above you consider F_c between 20 and 40 MPa.

Response R2.24: The authors appreciate the reviewer pointing this out. To address this comment, we have updated the model to use the relationship between the E-modulus for concrete and the compressive strengths per the design codes. (Page S14, below eq. 41)

Comment R2.25: Page S18. "for every kg of cement used, there are on average 7.3 kg of cement-based materials". This means 325kg of cement per cubic meter of concrete. Right? So you have the total cement production/consumption, you divide this amount per 325kg/m³ and determine the volume of concrete produced by that amount of cement. If my thinking is correct, you should rewrite the equation to improve the readability. If I did not understand, please clarify because is a little bit confusing. Verify even if eq 59 needs to be rewritten.

Response R2.25: The authors would like to thank the reviewer for this note. We have provided additional context after this clause. Namely, we state "This is to say, for every 1kg of cement used, there are approximately 6.3 kg of other constituents (e.g., water, aggregates) used. For example, with a 2,400 kg/m³ concrete mixture, approximately 330 kg of cement and 2070 kg of other constituents are used." We have used this clarification to improve readability; however,

because we implement the equation as written, we have not modified the equation itself. (Page S18)

Comment R2.26: Table S5. “Mean” is it the mean of the median values?

Response R2.26: The authors would like to thank the reviewer for commenting on the use of “median” and “mean” in Table S5. The median refers to reductions being compared based on the median GHG emissions of the baseline case and the four mitigation methods. The “mean” refers to the arithmetic mean reduction value of the three strength categories. We have added a clarification to the Table S.5 caption. (Page S20)

Comment R2.27: Page S25 item S.6.6. This analysis is preliminary. As authors claim, many aspects related to structural modeling have not been considered. Besides foundations, roofs etc., depending on the height of the building other structural elements are necessary. So, I think that the note at the second paragraph should be moved to the 1st paragraph and clearly state the limitation.

Response R2.27: The authors agree with the reviewer’s note and have moved the note regarding the limitation of the initial examination of the influence of building height to the first paragraph. (Page S26-S27)

Reviewer #3:

Comment R3.1: The authors present a detailed study of mitigation strategies not using still prospective technologies. Their paper is interesting, useful and well laid out. There are however some issues which have not been addressed which would help. Notably sensitivity studies are required for some parameters.

Response R3.1: The authors would like to the reviewer for taking the time to provide productive and detailed feedback on this paper.

Comment R3.2: The introductory paragraph sets the scene but it would be better to give a sense of the percentages, rather than "1 billion" which may or may not be a lot of concrete in relative terms. In particular, for their projections, the authors should consider a range of scenarios, in terms of population growth and urbanisation.

Response R3.2: The authors would like to thank the reviewer for this input. The percentage of urban population growth between 2018 and 2030 has been added (page 2, line 6-7). Further, the authors agree that a range of scenarios regarding the population growth and urbanization would add value to the analysis. To address this point, we conducted a new sensitivity analysis addressing variation in population growth. Please see page S26 for results from this sensitivity analysis.

Comment R3.3: The sentence "Critically, the efficient use of these materials must be a step in mitigating GHG emissions from material production" is cryptic. Do the authors mean that

geopolymers cannot ever abate emissions because the use resources that are used 1:1 in abating cement use and adds activators? They should say so clearly.

Response R3.3: The authors did not intend to claim that any material technology with lower GHG emissions from production than cement would not also be part of the suite of solutions. Rather, our goal was to highlight how lower consumption of materials in general is a key step to reducing burdens from materials production. We have rewritten the statement highlighted by the reviewer to read: “Critically, improving material efficiency, in which less material is used to achieve the same performance, is a key step in mitigating the environmental impacts from materials production. This step should be used in unison with low-emissions material alternatives to overcome GHG emissions challenges from the built environment.” (Page 3, line 3-5)

Comment R3.4: p 3 There are also alternative structural systems which have been proposed and which could be more efficient. In general, missing from the study is the impact of e.g. using more steel frames instead of concrete frames where possible.

Response R3.4: The authors appreciate the reviewer's input regarding alternative structural systems. While the authors agree that reductions can be achieved by using steel frames, instead of reinforced concrete in some cases, analysis of other materials than reinforced concrete is outside the scope of this work. Additionally, analysis of reinforced concrete versus steel frames is highly depending on the structural system under consideration and the results vary from case to case. We have clarified this in the text, see page 3, lines 20-24. [10.1061/\(ASCE\)1076-0342\(2005\)11:2\(93\)](https://doi.org/10.1061/(ASCE)1076-0342(2005)11:2(93))

Comment R3.5: p 5 The authors should compare tailoring mixes to requirements with the number of mixes already used on-site. Are they proposing to make construction much more complex?

Response R3.5: The authors understand the reviewer’s note, and we agree that tailoring mixes to specific requirement could indeed make construction more complex. We have clarified that tailoring concrete mixtures might not be feasible from a constructability, or economic standpoint for all applications on page 6, line 25 and page 7 line 1.

Comment R3.6: p 6 There can be other considerations, in particular acoustic comfort which may impose a minimum depth for the slab. Have the authors looked at the scope for optimisation assuming a sensible range of slab depths?

Response R3.6: The authors appreciate the reviewer taking the time to list additional considerations that may affect slab depth. For this analysis, the minimum slab thickness to not exceed allowable deflection limits for the specified span length has been considered. Regarding acoustic comfort, this consideration was excluded from the scope due to the fact that deflection generally is the governing factor in slab design. Here, our work prioritized meeting structural design requirements, and parameters such as acoustic performance among others could be considered in future work. (Page 7, line 22-23)

Comment R3.7: p 7 are the authors taking into consideration punching shear design? thicker

slabs may require less shear reinforcement...

Response R3.7: The authors appreciate the reviewer's concern regarding punching shear. However, punching shear or shear reinforcement is not considered in this analysis; we consider those aspects of modeling outside the scope of analysis, and here our focus is only including design for bending. In addition to shear reinforcement by the support, a variety of other options such as column caps could be considered as well (which would contribute to additional concrete volume). The authors recognize that additional structural design aspects besides those considered in this study can affect the ultimate environmental impact. We make a note of these limitations in the manuscript. (Page 17, line 21-22 and Page 18, line 8-9)

Comment R3.8: p 8 the authors need to more critically look at the reasons for demolitions. It is very uncommon that buildings be demolished due to reaching the end of their service life. much more common that they reached the end of their economic lives. The authors should do a sensitivity analysis looking at the impact of their proposed measure as a function of the effective lives of buildings (40-80 years)

Response R3.8: The authors appreciate the reviewer's input regarding the service life of structures and the reason for their demolition. We agree that the primary reason for buildings and infrastructure being taken out of service is not always due to poor durability (i.e., reaching the end of their structurally functional life). We have added a clarification to address this point. Regarding the modeling sensitivity analysis, we are modeling the cement used in buildings and infrastructure as not taken out of service at a certain year, but gradually as a function of a range of service lives, and hence the model is already capturing the variation in effective service life. We have improved the clarity of how service life has been modeled by adding a statement referring to service-life modeling being based on statistical analysis of longevity of concrete systems in 10 regions that cumulatively represent the world, as well as added a section about the service life and cement demand projection model to the Methods section in the manuscript. (Page 11 (line 14-16), page 14-15, and SI, page S20)

Comment R3.9: p 12 The authors should also consider the variation in cement amount in mix designs in "real life", to add error bars to their analysis...

Response R3.9: The authors agree that individual mixtures will have varying cement content. Some aspects of variation are plotted in Figure 2; however, to ensure readability of subsequent plots global averages are taken for the baseline.

REVIEWERS' COMMENTS

Reviewer #2 (Remarks to the Author):

The reviewed version is clearer and, in my opinion, can be accepted for publication

Reviewer #3 (Remarks to the Author):

I think the authors for having addressed my concerns. I believe the newly added sensitivity analysis helps adding robustness to the author's conclusions.

This paper should be accepted for publication.

Title: Near-term pathways for decarbonizing global concrete production
Authors: Josefine Olsson, Sabbie A. Miller, Mark G. Alexander
Manuscript Number: NCOMMS-23-03837A

REVIEWER COMMENTS

Reviewer #2:

Comment R2.1: The reviewed version is clearer and, in my opinion, can be accepted for publication.

Response R2.1: The authors greatly appreciate the reviewer's time. We are pleased to read that we have improved upon the original manuscript.

Reviewer #3:

Comment R3.1: I think the authors for having addressed my concerns. I believe the newly added sensitivity analysis helps adding robustness to the author's conclusions. This paper should be accepted for publication.

Response R3.1: The authors greatly appreciate the reviewer's time. We are pleased that our revisions have addressed the reviewer's earlier points.